# Validity of Calculating Continuous Relative Phase during Cycling from Measures Taken with Skin-Mounted Electro-Goniometers [note 1]

**DOI:** 10.3390/s22124371

**Published:** 2022-06-09

**Authors:** Chris Whittle, Simon A. Jobson, Neal Smith

**Affiliations:** 1School of Sport, Health and Community, University of Winchester, Winchester SO22 4NR, UK; simon.jobson@winchester.ac.uk; 2Chichester Institute of Sport, University of Chichester, Chichester PO19 6PE, UK; n.smith@chi.ac.uk

**Keywords:** electro-goniometers, validity, continuous relative phase, cycling

## Abstract

The aim of this study was to assess the validity of electro-goniometers as a tool for recording continuous relative phase data at two joint couplings during cycling tasks at a range of cadences. Seven participants (4 male, 3 female, age: 29 ± 7 years, height: 1.76 ± 0.10 m, mass: 71.97 ± 11.57 kg) performed exercise bouts of 30 s at four prescribed cadences (60, 80, 100, 120 rev·min^−1^) on a stationary ergometer (Wattbike, Nottingham, UK). Measures were synchronously recorded by bi-axial electro-goniometers (Biometrics, UK) and a 12-camera motion-capture system (Qualisys, Gothenburg, Sweden), with both systems sampling at 500 Hz. Sagittal plane joint angle and joint angular velocity were recorded at the hip, knee and ankle and analysed for ten complete pedal revolutions per participant per condition. Data were interpolated to 100 time points and used to calculate mean continuous relative phase (CRP) per pedal revolution at two intra-limb couplings: (i) knee flexion/extension–ankle plantarflexion/dorsiflexion (KA) and (ii) hip flexion/extension–knee flexion/extension (HK). At the KA coupling, significant differences in mean CRP were found between measurement systems at 120 rev·min^−1^ (*p* = 0.006). At the HK coupling, significant differences in mean CRP were found between measurement systems at 80 rev·min^−1^ (*p* = 0.043) and 100 rev·min^−1^ (*p* = 0.028). ICC values for most comparisons were below 0.5, suggesting poor levels of agreement between systems. Significant differences in mean CRP per pedal revolution and poor levels of agreement between systems suggests that electro-goniometers are not a suitable alternative to motion-capture systems when attempting to record CRP during cycling.

## 1. Introduction

Historically, cycling kinematics research has tracked joint and segment positions in an effort to calculate joint ranges of motion [1]. These joints are then, most commonly, analysed in isolation [2,3,4,5]. Although this is the most widely replicated approach, it has been criticised for not effectively capturing the complexity of coordinated motion [6].

As an alternative, it has been suggested that the continuous, multi-joint nature of the cycling task [7] lends itself best to a continuous relative phase (CRP) method of analysis, whereby the influence of one segment’s motion upon an adjacent segment can be more readily acknowledged. This is achieved by calculating the joint angle at each joint across the entire motion cycle and then using angle–angle plots. These plots can then be quantified using vector coding techniques to establish the relative motion of two adjacent joints [8].

CRP values can range from 0° to 360°, where 0° shows the respective movements of the coupled joints perfectly in-phase, and 180° indicates that they are perfectly anti-phase. Any value between these indicates a relative amount of in-phase or anti-phase movement.

Inconsistencies with this reporting convention have been identified [9], with some authors choosing to report values only between 0° and 180°, given that the values −180° and 180° both indicate anti-phase behaviour, whilst others utilise both the positive and negative values because they have qualitative meaning that should be preserved. For example, it has been suggested that preserving the negative values is important because if the phase angle of the proximal segment is subtracted from the phase angle of the distal segment, then positive continuous relative phase values indicate that the distal segment is ahead of the proximal segment in phase space, therefore providing a clearer image of the coupling’s interaction [10].

The level of detail offered by CRP analysis allows a more detailed evaluation of the interactions along the kinematic chain and has been suggested to be especially important where one end of the segmental chain is effectively fixed, in the case of cycling through its attachment to the pedal. The consideration of the coupling relationship between segments has been therefore suggested to be especially crucial in the analysis of cycling motion [11]. Additionally, CRP analysis has been deemed to be more sensitive to changes in coordination [12] and could offer greater insight into the changing techniques employed in response to learning environmental changes such as wind speed or road surface or other independent variables [13].

CRP has traditionally been measured using motion-capture systems in a laboratory setting [14,15,16]. This requires the duplication of a cyclist’s equipment using an ergometer due to the amount of distance covered during a cycling bout and the inability to calibrate such an extensive capture volume for kinematic analysis. There is, however, a readily available body of literature that focusses on the lack of ecological validity of such an approach. Studies have shown that there is a significant difference in cycling speed and power output between laboratory and road conditions during time trial events [17,18], whilst others have shown that crank torque profiles are significantly different when comparing laboratory and outdoor cycling conditions [19]. This has prompted calls to move towards a testing environment where riders can use their own bikes to accurately replicate ‘‘real-world’’ performance [1], an approach which may be facilitated by the use of electro-goniometers during field testing.

Electro-goniometers have long been used for the measurement of lower extremity joint motion [20], and their physical characteristics make them suitable for practical applications within biomechanics [21]. The lightweight equipment and non-invasive methods of data collection, coupled with the ability to record offline data logging systems, makes them a potentially excellent choice for field-based assessments within cycling. Indeed, they have already been assessed in terms of their suitability for use in professional bike-fitting services [22] and have been found to be more accurate and valid for use within laboratory studies than manual methods of measuring knee joint range of motion [23]. Despite this, to the best of the authors’ knowledge, electro-goniometers have yet to be used to calculate CRP during cycling efforts.

The aim of this study, therefore, was to extend the initial findings reported at the ECSS 25th Annual congress [24] in an effort to investigate whether electro-goniometers offer a valid method for the calculation of CRP values during cycling performance. If this is the case, investigations into cycling techniques can move to a more ecologically valid setting, whilst considering the interconnected nature of joint movements which occur during the movement.

## 2. Materials and Methods

### 2.1. Participants

Seven participants (4 male, 3 female, age: 29 ± 7 years, height: 1.76 ± 0.10 m, mass: 71.97 ± 11.57 kg) volunteered to take part in the study. Participants were recreationally active and free from injury at the time of testing but were not trained cyclists. All participants provided written informed consent before taking part in this study, which had local ethics committee approval in accordance with the rules of the Declaration of Helsinki of 1975, revised in 2013.

### 2.2. Procedure

Participants were invited to adjust the cycle ergometer (Wattbike Pro cycle ergometer, Wattbike, UK) to their comfort. This configuration was maintained throughout the testing session. Reflective markers (Qualisys, Sweden) were attached to the participant’s right leg at the greater trochanter, lateral femoral condyle and lateral malleolus. A marker was also attached to the lateral side of the participant’s shoe, with placement determined by palpation to establish the positioning of the base of the 5th metatarsal. Bi-axial electro-goniometers (Biometrics, UK) were attached at the hip, knee and ankle. The electro-goniometer at the hip was aligned vertically with the strain gauge running immediately posterior to the greater trochanter marker and the terminals positioned equidistant superior and inferior to the marker. The electro-goniometer at the knee was positioned on the medial aspect of the knee, aligned vertically with the strain gauge running directly over the medial femoral condyle and the terminals equidistant superior and inferior to this landmark. The electro-goniometer at the ankle was attached so that the superior terminal was aligned vertically above the medial malleolus, the strain gauge ran over the medial malleolus and the inferior terminal was positioned horizontally on the participant’s shoe so that the electro-goniometer recorded an angle of 90° with the participant standing in the anatomical reference position. Goniometers were “zeroed” before application and applied to achieve values close to 0°, 0° and 90°, respectively.

Participants performed exercise bouts of 30 s at four prescribed cadences (60, 80, 100, 120 rev·min^−1^) on the stationary ergometer (Wattbike, UK), with freely chosen resistance. Participants were given free choice of riding posture but asked to maintain the same position across all conditions.

### 2.3. Data Analysis

Measures were synchronously recorded by the bi-axial electro-goniometers (Biometrics, UK) and a 12-camera motion-capture system (Qualisys, Sweden), with both systems recording at 500 Hz. Raw marker trajectories were used to calculate sagittal plane joint angle and joint angular velocity, which were recorded at the hip, knee and ankle and analysed for 10 complete pedal revolutions per participant per condition. Data were interpolated to 100 time points and used to calculate mean continuous relative phase (CRP) per pedal revolution at two intra-limb couplings: (i) knee flexion/extension–ankle plantarflexion/dorsiflexion (KA) and (ii) hip flexion/extension–knee flexion/extension (HK).

Following checks for normal distribution, a combination of repeated measures T-tests and Wilcoxon signed rank tests were used to check for significant differences between measurement systems, followed by intra-class correlation coefficients (ICC) via the two-way mixed model to quantify the consistency of the CRP values produced by the two systems.

All statistical testing was performed using IBM SPSS statistics (IMB Corporation, Armonk, NY, USA), with an alpha level set at *p* < 0.05.

## 3. Results

When comparing the mean CRP values produced by the two systems (Table 1), there were statistically significant differences (*p* < 0.05) at 80 and 100 rev·min^−1^ for the Hip–Knee coupling and at 120 rev·min^−1^ for the Knee–Ankle coupling.

The goniometers appeared to report consistently higher mean values at the Hip–Knee coupling across all cadences. This is also true for 80, 100 and 120 rev·min^−1^ for the Knee–Ankle coupling, with the goniometers apparently under-reporting at 60 rev·min^−1^, compared to the previously validated camera system (Table 1).

Intra-class correlation coefficients were created via the two-way mixed model to quantify the consistency of the CRP values produced by the two systems (see Table 1). The majority of these coefficients were below 0.5, suggesting poor levels of reliability between systems. The only exceptions to this were seen at 80 and 100 rev·min^−1^ at the Knee–Ankle coupling, where values of 0.749 and 0.664, respectively, were recorded. This would suggest, at best, a moderate level of agreement between systems, and predicated further investigation into the basic joint position data produced by each system to ascertain the reason for such discrepancies.

Comparing positional data between systems using Wilcoxon signed rank tests, it became apparent that there were significant differences (*p* < 0.05) at all cadences when comparing mean maximum hip angle and mean minimum hip angle (Table 2). The only exception to this was at 80 rev·min^−1^ (*p* = 0.197), where there was no statistically significant difference between the two systems; however, the large standard deviation value (±18.95) in the goniometer dataset does offer some cause for concern.

When comparing the mean maximum knee angle, there was further evidence that the two systems did not agree, with statistically significant differences (*p* < 0.05) being seen at all cadences (see Table 3). This was also the case when comparing the mean minimum knee angle (see Table 3). Again, statistically significant differences (*p* < 0.05) were recorded at all cadences.

Levels of reported ankle flexion/extension were also statistically significantly different (*p* < 0.05) between the two measurement systems at all cadences with regards to both maximum and minimum mean reported values (see Table 4).

In summary, positional data suggested that the goniometer systems consistently over-reported both maximum and minimum values for hip and knee flexion/extension, while simultaneously under-reporting the corresponding values at the ankle.

## 4. Discussion

Results from this investigation suggest that bi-axial electro-goniometers are not a valid method for recording CRP values during simulated cycling efforts. There were statistically significant differences (*p* < 0.05) between measurement systems in two of four tested cadences for the Hip–Knee coupling, and a further significant difference was reported at 120 rev·min^−1^ for the Knee–Ankle coupling. The lack of agreement between systems was further supported by ICC values, which mostly fell below 0.5, showing poor levels of agreement between systems [25] when calculating CRP.

The discrepancy between systems could be because signal values were not normalised. There has been some debate as to whether or not normalisation would avoid the magnitude of values from one segment dominating the CRP pattern [9]. However, multiple studies [9,10] concluded that, in the case of joint kinematics, normalisation is not required because the finite values are unimportant—it is the relative phase that is of interest. Calculation of CRP, therefore, appears to require normalisation of values against time, as performed here, but not normalisation of the original signal values themselves.

As shown above, further investigation into the reason for the lack of agreement revealed statistically significant differences (*p* < 0.05) between systems at the fundamental level of measured angular position. The two systems only agreed in terms of the minimum angle recorded at one joint (the hip), in one condition (80 rev·min^−1^). All other comparisons returned significantly different results. Discrepancies at this level make it almost inevitable that there will be differences between reported CRP values, based, as they are, on differing fundamental measures.

The reason for such discrepancies in basic measures of angular position could, in part, be attributed to poor experimental control in terms of goniometer placement. Although every effort was made to replicate the exact placement described in the Methods Section above, the lack of anatomical landmarks to use for reference means it is possible that there was some variation in placement between participants.

Even if placement was perfectly replicated between participants, it has been suggested that the human body lacks even surfaces and right angles on which to attach sensors of this nature to accurately calculate joint angles [26]. The suggestion being that the lack of flat surfaces means the orientation of a measurement device cannot possibly be aligned with any physiologically meaningful axis. This is especially apparent at the knee, where despite traditionally being described as a single planar hinge joint, there are degrees of freedom relating to flexion/extension, abduction/adduction and internal/external rotation [27,28]. Although abduction/adduction and internal/external rotation angles very rarely exceed a range of ±10° [29], it is possible that this is enough to affect the measurement of angular position when using a system such as the electro-goniometers used here, which assume entirely planar motion.

Related concerns with the placement of the electro-goniometers include the influence of soft-tissue movement artifacts, the suggestion that surface-mounted markers may not adequately represent true anatomical locations and the assumption that markers attached to the skin surface are rigidly connected to the underlying bones [30,31]. It has been reported that skin marker trajectories showed up to a 31 mm error, when compared to a prosthesis-embedded anatomical frame, and up to a 192% root mean square error in abduction/adduction estimations taken from markers placed on the thigh and shank. Although the reflective markers used in this investigation were placed on bony anatomical landmarks (greater trochanter, lateral femoral condyle and lateral malleolus) to remove the influence of such artifacts, it should be noted that it is not possible to mount the electro-goniometers in such a way. The electro-goniometers, therefore, may have been subject to the type of soft-tissue movement artifacts described above, and this could contribute to the lack of agreement between systems in terms of fundamental angular position and CRP.

A potential limitation of the current study relates to the way in which the measures were produced. Although care was taken to match the sampling frequencies of the systems at 500 Hz and the same 10 revolutions were analysed per participant per condition, the systems themselves were not synchronised. It is possible that this may have contributed to the differences seen between systems, but it is worth noting that, even at the highest cadence (120 rev·min^−1^), the chosen sampling rate still provides approximately 250 measures per pedal revolution.

In the current investigation, CRP was reported as a mean value for an entire pedal revolution. The poor agreement between systems shown at this level meant that it was deemed more worthwhile to investigate the root of the discrepancies between systems rather than delve further into the divisions of a pedal revolution, but this is something which would be recommended once a valid measurement system has been established. Reporting a single CRP value averaged across a complete pedal revolution may not offer enough detail throughout the various phases of the revolution to fully exhibit the nuanced kinematics at play. Therefore, it is suggested that future studies should split the pedal revolution into separate power and recovery phases. This approach has been adopted previously [32] and has, at times, been extended to an even more detailed analysis of four “quarters” across the pedal revolution [33,34,35]. The purpose of such a split would be to effectively separate the power and recovery phases from the areas at the top and bottom of the pedal revolution, which have long been identified as areas where pedalling kinematics are altered due to tangential force being at a minimum [36,37].

## 5. Conclusions

Although it has been suggested that the use of CRP analysis provides information that cannot be obtained through conventional angular position vs. time presentation, the results from this study would suggest that bi-axial electro-goniometers are not a suitable method for recording such values.

Further investigations are recommended to establish a valid alternative to traditional motion-capture systems so that investigations into joint-couple motions during cycling may move to a more ecologically valid setting that accurately replicates the “real-world” performances of athletes.

## Figures and Tables

**Table 1 sensors-22-04371-t001:** Comparisons between mean continuous relative phase values produced across a complete pedal revolution.

Coupling	Cadence (rev·min^−1^)	Mean CRP Value (Mean ± SD)		Sig.	ICC
		Camera System	Goniometers		
Hip–Knee	60	3.57 (±1.94)	5.55 (±1.05)	0.080	−0.413
Hip–Knee	80	3.33 (±2.36)	6.81 (±1.84)	0.043 *	−0.272
Hip–Knee	100	2.48 (±1.76)	7.19 (±1.73)	0.028 *	−0.103
Hip–Knee	120	7.81 (±6.57)	13.59 (±5.23)	0.191	−0.418
Knee–Ankle	60	11.43 (±4.83)	8.71 (±3.36)	0.066	0.749
Knee–Ankle	80	12.31 (±6.13)	13.17 (±6.67)	0.691	0.664
Knee–Ankle	100	12.26 (±6.70)	18.95 (±13.11)	0.176	0.346
Knee–Ankle	120	11.29 (±5.10)	29.22 (±16.25)	0.009 *	0.376

* Denotes a significant difference between systems at *p* < 0.05.

**Table 2 sensors-22-04371-t002:** Comparison of mean maximum and mean minimum hip angle recorded across 10 pedal revolutions.

**Cadence (rev·min^−1^)**	60		80		100		120	
**Measurement System**	Camera	Goniometer	Camera	Goniometer	Camera	Goniometer	Camera	Goniometer
**Maximum Hip Angle (^o^)**	73.25 (±2.10)	84.08 (±13.70)	73.56 (±2.00)	82.22 (±17.30)	73.37 (±2.42)	82.88 (±15.85)	71.80 (±2.75)	83.52 (±16.89)
**Sig.**	<0.001 *		<0.001 *		<0.001 *		<0.001 *	
**Minimum Hip Angle (^o^)**	33.49 (±5.21)	40.79 (±17.71)	33.87 (±5.65)	36.30 (±18.95)	33.21 (±5.60)	37.11 (±19.25)	31.02 (±5.92)	39.24 (±17.70)
**Sig.**	0.010 *		0.197		0.044 *		<0.001 *	

* Denotes a significant difference between systems at *p* < 0.05.

**Table 3 sensors-22-04371-t003:** Comparison of mean maximum and mean minimum knee angle recorded across 10 pedal revolutions.

**Cadence (rev·min^−1^)**	60		80		100		120	
**Measurement System**	Camera	Goniometer	Camera	Goniometer	Camera	Goniometer	Camera	Goniometer
**Maximum Knee Angle (^o^)**	138.75 (±8.66)	165.24 (±6.36)	138.52 (±9.39)	166.99 (±6.07)	138.61 (±8.87)	170.04 (±5.36)	140.26 (±9.74)	173.62 (±8.19)
**Sig.**	<0.001 *		<0.001 *		<0.001 *		<0.001 *	
**Minimum Knee Angle (^o^)**	70.75 (±4.17)	113.25 (±13.35)	70.42 (±4.44)	116.62 (±14.08)	69.81 (±4.40)	117.41 (±13.29)	70.17 (±4.92)	121.00 (±15.70)
**Sig.**	<0.001 *		<0.001 *		<0.001 *		<0.001 *	

* Denotes a significant difference between systems at *p* < 0.05.

**Table 4 sensors-22-04371-t004:** Comparison of mean maximum and mean minimum ankle angle recorded across 10 pedal revolutions.

**Cadence (rev·min^−1^)**	60		80		100		120	
**Measurement System**	Camera	Goniometer	Camera	Goniometer	Camera	Goniometer	Camera	Goniometer
**Maximum Ankle Angle (^o^)**	120.65 (±11.98)	102.31 (±9.61)	117.97 (±5.67)	102.18 (±8.70)	118.11 (±6.15)	104.41 (±13.20)	119.68 (±5.31)	114.49 (±48.72)
**Sig.**	<0.001 *		<0.001 *		<0.001 *		<0.001 *	
**Minimum Ankle Angle (^o^)**	100.90 (±13.35)	83.59 (±7.17)	95.39 (±7.38)	83.23 (±7.17)	94.91 (±6.92)	83.23 (±6.80)	94.80 (±5.26)	79.22 (±12.10)
**Sig.**	<0.001 *		<0.001 *		<0.001 *		<0.001 *	

* Denotes a significant difference between systems at *p* < 0.05.

## Data Availability

The data presented in this study are available on request from the corresponding author. The data are not publicly available due to the assurance of anonymity given to all participants prior to their involvement in the study.

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
