# Peer review of "Validity of Calculating Continuous Relative Phase during Cycling from Measures Taken with Skin-Mounted Electro-Goniometers†"

_sensors, 2022, doi:10.3390/s22124371_

Round 1

Reviewer 1 Report

It seems electro-goniometers are also considered one of standard devices. Why does it have to be validated?

Sample size is very small. It must be justified.

Data analysis plan is not well described. What were statistical analyses for comparing two different systems?

Some of core analyses for comparing two different systems such as Bland-Altman plot has not been done.

Author Response

  • It seems electro-goniometers are also considered one of standard devices. Why does it have to be validated?

Thank you for your comment. An additional sentence has been added to the introduction (line 85) to emphasise this point.

  • Sample size is very small. It must be justified.

Although we acknowledge that there are only 7 participants the decision to analyse 10 full revolutions per participant ensures that the analysis of joint angles was conducted on 70 comparisons between systems in each cadence condition. Across the 4 conditions this means that a total of 280 comparisons have been conducted between systems. 70 data points provides, in our opinion, more than enough data to be confident in the statistical analysis performed. Morse (1999) cited Kirk (1990) as suggesting that Wilcoxon Signed Rank Tests give “satisfactory accuracy with sample sizes of 10 or more”.

  • Data analysis plan is not well described. What were statistical analyses for comparing two different systems?

Initial comparisons of CRP values between the two systems were tested via T-tests and Wilcoxon Signed Rank Tests (depending on the normality of distribution). This was followed by interclass correlation coefficients. The resulting poor levels of agreement led to additional comparisons of the raw joint position data between systems. This is explained in the method (line 133) and the results section (line 156) has been edited to clarify.  

  • Some of core analyses for comparing two different systems such as Bland-Altman plot has not been done.

Bland-Altman plots were not initially included as the results from the interclass correlation coefficients were deemed to be sufficient evidence of the poor agreement between systems. These have, however, now been completed and representative examples are attached here for reference. These can be refined and included in the article if the reviewer/editor believes that doing so would help with clarity.

Reviewer 2 Report

This work was to address the validity of electro-goniometers in CRP (continuous relative phase) analysis. I have the following comments:

1) In Introduction, I wonder what is the major goal of this work. If you want to emphasize that  the CRP analysis is crucial in cycling performance evaluation, then you should find a workable method to offer reliable and correct results, rather than proving that the electro-goniometers is not as good as expected. On the other hand, if the goal is to prove that the electro-goniometer is not suitable for CRP analysis, then more detailed and  sophisticated tests and results should be provided. Currently, the authors listed several possible reasons or limitations about the invalidity of the   electro-goniometers in Discussion. More complete investigation should be designed and provided. 

2) Only seven participants joined the tests. I wonder the data amount is enough for doing the statisitcs, such as T-tests. 

3) The comparison was done by comparing the electro-goniometer only with the camera system. It is improper. Other wearable sensors should be considered in the comparison. 

4) This work is an extended version of the author's previous study. An explicit comparison or explanation of the differences is necessary. 

Author Response

  • In Introduction, I wonder what is the major goal of this work. If you want to emphasize that the CRP analysis is crucial in cycling performance evaluation, then you should find a workable method to offer reliable and correct results, rather than proving that the electro-goniometers is not as good as expected. On the other hand, if the goal is to prove that the electro-goniometer is not suitable for CRP analysis, then more detailed and sophisticated tests and results should be provided. Currently, the authors listed several possible reasons or limitations about the invalidity of the   electro-goniometers in Discussion. More complete investigation should be designed and provided. 

Thank you for the feedback. The introduction has had an additional sentence added (line 85) to clarify that the aim of this investigation was to ascertain whether electro-goniometers offer a valid alternative to motion capture systems when trying to calculate continuous relative phase during cycling performance.

  • Only seven participants joined the tests. I wonder the data amount is enough for doing the statistics, such as T-tests. 

Although we acknowledge that there are only 7 participants the decision to analyse 10 full revolutions per participant ensures that the analysis of joint angles was conducted on 70 comparisons between systems in each cadence condition. Across the 4 conditions this means that a total of 280 comparisons have been conducted between systems. 70 data points provides, in our opinion, more than enough data to be confident in the statistical analysis performed. Morse (1999) cited Kirk (1990) as suggesting that Wilcoxon Signed Rank Tests give “satisfactory accuracy with sample sizes of 10 or more”.

  • The comparison was done by comparing the electro-goniometer only with the camera system. It is improper. Other wearable sensors should be considered in the comparison. 

Although we thank you for the feedback, in this case we must respectfully disagree. The aim of this investigation was to ascertain whether electro-goniometers offer a valid alternative to motion capture systems when trying to calculate continuous relative phase during cycling performance. The gold standard of 3D motion capture is not available outdoors in an ecologically valid setting, and the main problem with many commercial “sensor” solutions is the lack data storage capacity when used outside of internet connectivity or bluetooth transmissions ranges. It is, therefore, not improper to compare electro-goniometers against the gold standard as done here and the inclusion of other wearable sensors would fall beyond the remit of this investigation.

  • This work is an extended version of the author's previous study. An explicit comparison or explanation of the differences is necessary. 

Again, we thank you for the feedback but have already discussed this with the editor and have been assured that the existing explanation and the fact that this manuscript is being presented as a conference extension will meet the requirements of the journal.

Round 2

Reviewer 1 Report

Thank you for addressing my comments. I have no more comments.

Reviewer 2 Report

I have no any comments.